# Photocatalytic Multicomponent Annulation of Amide-Anchored 1,7-Diynes Enabled by Deconstruction of Bromotrichloromethane

**DOI:** 10.3390/molecules29040782

**Published:** 2024-02-08

**Authors:** Daixiang Chen, Yu Bao, Shenghu Yan, Jiayin Wang, Yue Zhang, Guigen Li

**Affiliations:** 1School of Pharmacy, Changzhou University, Changzhou 213164, China; 2Department of Chemistry and Biochemistry, Texas Tech University, Lubbock, TX 79409, USA; guigen.li@ttu.edu

**Keywords:** C1 chemistry, multicomponent annulation, 1,7-diynes, photocatalytic, Kharasch-type addition

## Abstract

We present the first example of visible-light-mediated multicomponent annulation of 1,7-diynes by taking advantage of quadruple cleavage olf carbon-halogen bonds of BrCCl_3_ to generate a C1 synthon, which was adeptly applied to the preparation of skeletally diverse 3-benzoyl-quinolin-2(1*H*)-one acetates in moderate to good yields. Controlled experiments demonstrated that H_2_O acted as both oxygen and hydrogen sources, and *gem*-dichlorovinyl carbonyl compound exhibited as a critical intermediate in this process. The mechanistic pathway involves Kharasch-type addition/6-exo-dig cyclization/1,5-(SN”)-substitution/elimination/binucleophilic 1,6-addition/proton transfer/tautomerization sequence.

## 1. Introduction

C1 chemistry has emerged as a flourishing approach for the formation of homologous compounds, which introduced an additional carbon atom into the substrates to assemble valuable molecular frameworks [1,2,3]. C1 chemistry has gained significant attention in the field of organic synthesis due to its ability to increase the length of carbon chains [4] and construct important functional groups (e.g., carboxylic or carbonyl groups) [5], which was utilized in the modification of pharmaceuticals, agrochemicals and bioactive natural products for value addition [6,7]. As we all know, CO_2_, CO and HCOOH are useful one-carbon building moiety, which have been widely exploited in many interesting and unprecedented transformations [8]. In the past few years, Song and coworkers discovered that several readily available halodifluoroalkyl compounds, including BrCF_2_COOEt, ClCF_2_COONa and ClCF_2_H, could act as a C1 precursor via the activation of two aliphatic C–F bonds along with the cleavage of C-X (X = Cl, Br or I) and C-R bonds, that is, a quadruple cleavage on one molecule in one pot [9,10,11,12,13].

Compared with halodifluoromethyl reagents [14], CBrCl_3_ or CBr_4_ is also widely pursued as a catalyst, reagent, and mediator for the preparation of polyhalogenated compounds in various organic transformations [15,16]. Among them, these tetrahalomethanes can undergo homolytic cleavage to deliver a bromine radical and a trihalomethyl radical by using visible light irradiation, which provides a green and efficient method for preparing important *gem*-dihaloalkene scaffolds through double cleavage of C-Br and C-X (X = Cl, Br) bonds (Figure 1a, path I) [17,18,19,20]. In contrast to the well-developed methodology for building up *gem*-dihaloalkenes by means of tetrahalomethanes as radical precursor, investigations on bromotrichloromethane serving as C1 sources via quadruple cleavage are considerably less common (Figure 1a, path II). For this reason, developing new, practical and sustainable photocatalytic methods that enables the direct deconstruction of tetrahalomethanes to add one extra carbon into functionalized organic molecules remains a formidable challenge, which might have deep impact on C1 chemistry as well.

Visible light-induced photoredox catalysis (VLPC) has been confirmed as one of the most attractive and powerful protocols for initiating various radical chemistry to construct a plethora of complex organic molecules because of its intrinsic characteristics of greenness, availability, safety as well as sustainability [21,22,23,24,25,26,27,28]. This type of reaction provides a unique and direct way to construct intricate architectures that cannot be obtained by thermal processes [29,30,31,32,33,34,35]. In particular, photocatalytic annulation cascade reactions of 1,*n*-diynes appear as a remarkable and applicable platform for facilitating the formation of poly-substituted isocyclic and heterocyclic compounds via synergistic interactions across their two unsaturated carbon–carbon triple bond [36,37,38]. Furthermore, Kharasch-type addition/cyclization of unsaturated systems by taking advantage of photocatalytic strategy is a convenient and atom economy method for producing polyhalogenated products [39,40,41,42,43]. For instance, the group of Jiang elaborated a photocatalytic three-component biheterocyclization of heteroatom-linked 1,7-diynes with CBrCl_3_ and water, leading to access skeletally diverse fused-tricyclic heterocycles [38]. Encouraged by previous results and the continuation of our interest in synthetic chemistry [44,45,46,47], we planned and conceived that the reaction can be directed toward photoinduced addition-annulation to build up quinolone skeletons when secondary alcohol-tethered 1,7-diynes was changed by amide-anchored 1,7-diynes as radical receptors. Intriguingly, we found the multicomponent cyclization reaction of amide-anchored 1,7-diynes **1** with CBrCl_3_, H_2_O and alcohols/thiols under visible-light irradiation, producing a plethora of 3-benzoyl-quinolin-2(1*H*)-one acetates **2** in moderate to excellent yields (Figure 1b). During this protocol, CBrCl_3_ serves as a C1 source by fully breaking one C–Br bond and three C–Cl bonds, and H_2_O acted as both oxygen and hydrogen sources in a single pot. To the best of our knowledge, this Kharasch-type multicomponent-annulation of amide-anchored 1,7-diynes via deconstruction of bromotrichloromethane remains unprecedented so far. Herein, we document these attractive transformations.

## 2. Results and Discussion

At the beginning, *N*-benzyl-*N*-(2-ethynylphenyl)-3-phenylpropiolamide (**1a**) and CBrCl_3_ were selected as the model substrates to establish reaction conditions, and the detailed results are shown in Table 1. The reaction of **1a** with CBrCl_3_ was irradiated under 30 W blue LEDs in the presence of *fac*-Ir(ppy)_3_ as photocatalyst in EtOH and H_2_O (*v*/*v*, 100:1) as co-solvent at 60 °C. The reaction proceeded poorly, and unexpected 3-benzoyl-quinolin-2(1*H*)-one acetate **2a** was observed in 48% yield after 36 h (Table 1, entry 1). The yield of product **2a** was increased to 67% in the presence of K_2_CO_3_ as base (Table 1, entry 2). Reducing reaction time will not improve the conversion of this transformation, as there is a large amount of starting material remaining (Table 1, entry 3). On the contrary, increasing reaction time led to reduced yields, but no side products were separated (Table 1, entry 4). The generation of **2a** was seriously inhibited in the absence of the *fac*-Ir(ppy)_3_ catalyst (Table 1, entry 5), suggesting that photocatalyst plays an important role in this process. Next, we examined other photocatalysts including [Ir(dFCF_3_ppy)_2_dtbbpy]PF_6_, Mes-Acr^+^ClO_4_^−^ and Eosin Y, and found that all of them exhibited poorer catalytic performance than that of *fac*-Ir(ppy)_3_ with respect to the yield of **2a** (Table 1, entries 6 and 8). Replacing K_2_CO_3_ as other inorganic bases including Na_2_CO_3_, Cs_2_CO_3_ and MeONa did not have any favorable impact on this reaction (entries 9–11). Similarly, Et_3_N and 4-dimethylaminopyridine (DMAP) as organic base also resulted in lower yields compared to that of K_2_CO_3_ (entries 12–13 vs. entry 2). Moreover, increasing or decreasing the amount of H_2_O in the co-solvent could not promote the efficiency of the multi-component reaction (entries 14–15). Finally, the use of CBr_4_ as a Cl precursor to replace CBrCl_3_ under standard conditions enabled a similar process to give rise to product **2a**, but with a significant drop in the yield (Table 1, entry 16).

Having established the optimal reaction conditions, we then evaluated the substrate scope and generality of various amide-linked 1,7-diynes and alcohols for this photocatalytic multicomponent deconstruction of CBrCl_3_, and the results are summarized in Figure 2. Firstly, ethanol as nucleophilic reagent reacted with 1,7-diynes **1** to investigate the influence of electronic properties and positions of substituents in the arylalkynyl units (R^1^), and all of them conveniently participated in the current multicomponent cyclization with acceptable yields. Both electron-donating (such as methyl **1b** and **1c**, methoxy **1d**, and *tert*-butyl **1e**) and electron-withdrawing (fluoro **1f**) groups located at the *meta*- or *para*-position of the arylalkynyl moiety can all work well in this system, affording the corresponding quinolin-2(1H)-ones **2b**–**2f** in 49–65% yields. However, the obvious impact on steric hindrance and electronic effect were demonstrated because arylalkynyl with *ortho* substituted (*o*-Me, *o*-Cl) or strong electron withdrawing groups (*p*-Br, *p*-CO_2_Me) inhibited the progress of the reaction, delivering nearly no desired product. Next, 1,7-diynes with different benzyl groups at the nitrogen atom have been proved to be suitable reaction partners under standard condition. The benzyl group bearing a functional group, such as ether (*o*-methoxy **1g**), alkyl (*p*-methyl **1j**) and halogen (*m*-fluoro **1h**, *m*-chloro **1i**, *p*-fluoro **1k**, *p*-chloro **1l**, *p*-bromo **1m**) proceeded smoothly, enabling their addition-cyclization to render the desired products **2g**–**2m** with yields ranging from 47 to 69%. Subsequently, we chose methyl (**1n** and **1o**) as the representative functional group to introduce C4 or C5 position of the internal arene ring of diynes to investigate its synthetic utility. The corresponding 3-benzoyl-quinolin-2(1*H*)-one acetates were isolated in 55% and 60% yields, respectively.

We next devoted our effort to explore the scope of this radical-triggered multicomponent annulation by taking advantage of different alcohols as nucleophiles. The compound 1,7-diynes **1a** was subjected to the treatment of various alcohols consisting of linear (methyl, but-3-yn-1-yl, benzyl), branched (isopropyl, isobutyl) and cyclic (cyclohexyl), which were proven to be applicable for this method, as targets **2p**–**2u** were generated in 51–65% yields. Promisingly, both ethanethiol and propane-2-thiol are good candidates, providing the quinolones-containing ethanethioate **2v**–**2x** in 63% and 71% yields, respectively. It is interesting and unexpected that thiols act as nucleophilic reagents and solvents, rather than as radical donors [48,49]. Unfortunately, *N*-Me protected **1p** or *N*-unprotected amide-linked 1,7-diyne **1q** and ester linked 1,7-diyne **1r** were not effective reaction partners, as they were unsuccessfully transformed into corresponding products. Furthermore, the preformed substrate **1s** with two internal alkyne moieties was examined, but the reaction did not proceed under this catalytic system and 1,7-diyne **1s** was recovered, showing that terminal alkynes on starting material play an important role for this transformation.

The gram-scale experiments for preparation of **2a** on a 4.0 mmol scale were conducted under optimal conditions, and the product was delivered with a comparable yield (60%, Figure 3a). Then, the practicality of this approach was further studied by the late-stage functionalization of products. For example, the treatment of **2a** with hydrazine hydrate in EtOH by using HOAc as acid, resulting in 3-amino-6-benzyl-4-phenylbenzo[*c*][2,7]naphthyridine-2,5(*3H*,*6H*)-dione **3** in 82% yield (Figure 3b).

Several control experiments were performed to gain mechanistic insight into the reaction pathway. Firstly, the use of a radical inhibitor TEMPO (2,2,6,6-tetramethyl-1-piperidinyloxy) successfully suppressed the reaction process, and the result confirmed that a trichloromethyl radical may be involved in these transformations (Figure 4a). When a 4 Å molecular sieve was employed as dehydrating agent and dry EtOH as solvent under standard conditions, the reaction process was completely inhibited (Figure 4b). In addition, the reaction was carried out in the presence of H_2_^18^O (98 atom% ^18^O), and the products **2a** and **2a**-O^18^ were identified by HR-MS in a 3:1 ratio (Figure 4c). These two survey results indicated that the oxygen source of the ester and carbonyl group in target products come from water. The deuterium-labeling experiment was performed to confirm the hydrogen source of methylene. In the presence of D_2_O, the deuterated product [D]-**2a** was obtained with the percentage content of deuterium being 69% (Figure 4d). The reaction between **1a** with CBrCl_3_ in wet-acetonitrile at room temperature in the presence of K_3_PO_4_ as base, which underwent Kharasch addition/1,5-(S_N_″)-substitution cascade process to form *gem*-dichlorovinyl carbonyls product **4** with 37% yield, followed by reaction with EtOH and H_2_O (*v*/*v*, 100:1) at 60 °C for 36 h, leading to the desired product **2a** in 75% yield. These observations suggest that *gem*-dichlorovinyl may serve as a key intermediate for the generation of product **2** (Figure 4e,f). Finally, several fluorescence quenching experiments indicated that CBrCl_3_ was a more efficient quencher of the excited state of *fac*-Ir(ppy)_3_* than 1,7-diyne **1a** (Figure 1, see the Appendix A for details).

Combined with these experimental outcomes and previous related works [17,18,19,20], we proposed a plausible mechanism about this photoredox-catalyzed multicomponent annulation of 1,7-diynes. As shown in Figure 5, the photocatalytic cycle was initiated by the activation of Ir(III) with blue light irradiation to form the excited state Ir(III)* species, which reacts with BrCCl_3_ to yield trichloromethyl radical **A** and a bromine anion, together with Ir(IV) complex via a single electron transfer (SET). Next, the radicals **A** can be trapped by the terminal carbon–-carbon triple bond of 1,7-diyne **1a** to provide the alkenyl radical **B**, which undergoes 6-*exo*-*dig*-cyclization to give intermediate **C**. The resulting bromine anion was oxidized by Ir(IV) complex to produce Br radical [50,51], followed by radical cross coupling with **C** to obtain intermediate **D** and to regenerate Ir(III) species. Subsequently, the intermediate **D** reacts with OH^−^ from H_2_O to afford the intermediate **E** through 1,5-(S_N_″)-substitution, which eliminates one molecule of HBr to assemble a key intermediate **4**. The intermolecular 1,6-addition of hydroxyl anion into intermediate **4** gives intermediate **F**, followed by protonation and removal of HCl to furnish intermediate **H**, which then undergoes nucleophilic 1,6-addition with ethanol, following by proton transfer (P.T.) and removal of HCl to generate intermediate **K**. Finally, the intermediate **K** transforms into the desired products **2a** through tautomerization.

## 3. Materials and Methods 

General procedure for the synthesis of compounds **2**

Example for the synthesis of **2a**:

In a 25-mL Schlenk tube, 1,7-diyne 1a (0.2 mmol, 67.0 mg, 1.0 equiv), BrCCl_3_ (0.4 mmol, 79.2 mg, 2.0 equiv), K_2_CO_3_ (0.4 mmol, 55.2 mg, 2.0 equiv), *fac*-Ir(ppy)_3_ (1.3 mg, 1 mol%) and EtOH/H_2_O (2.0 mL, 100:1 *v*/*v*) were successively added under Ar conditions. Then, the tube was stirred at 60 °C in oil bath for 36 h under 30 W blue light (blue LEDs) irradiation until complete consumption of 1a as monitored by TLC analysis. After the reaction was completed, the reaction mixture was concentrated in vacuum and the resulting residue was purified by column chromatography on silica gel (eluent, petroleum ether/ethyl acetate = 10:1) to afford the desired product 2a (57.0 mg, 67%) as a white solid.

General procedure for the synthesis of compounds **3**

To compound **2a** (0.1 mmol, 42.5 mg) a solution of hydrazine hydrate (0.5 mL) and acetic acid (one drop) in 1.0 mL EtOH was added and the mixture was stirred at 50 °C for 2 h until TLC analysis showed that 2a was completely consumed. The reaction mixture was then diluted with water, extracted with ethyl acetate, washed with water until neutral, and concentrated the organic layer under reduced pressure. The residue was purified through preparative thin layer chromatography (petroleum ether/ethyl acetate = 1:1 *v*/*v*) to afford compound **3** as white solid.

General procedure for the synthesis of compounds **4**

In a 10-mL Schlenk tube, 1,7-diyne 1a (0.1 mmol, 33.5 mg, 1.0 equiv), BrCCl_3_ (0.2 mmol, 39.6 mg, 2.0 equiv), Na_3_PO_4_ (0.2 mmol, 32.8 mg, 2.0 equiv), *fac*-Ir(ppy)_3_ (0.6 mg, 1 mol%) and MeCN (1.0 mL) were successively added under Ar conditions. Then, the tube was stirred at room temperature for 12 h under 30 W blue light (blue LEDs) irradiation until complete consumption of 1a as monitored by TLC analysis. After the reaction was completed, the reaction mixture was concentrated in vacuum and the resulting residue was purified by column chromatography on silica gel (eluent, petroleum ether/ethyl acetate = 15:1) to afford the desired product 4 (16.0 mg, 37%) as a white solid.

## 4. Conclusions

In summary, starting from easily prepared amide-anchored 1,7-diynes and available BrCCl_3_, we have discovered a novel photocatalytic carbon–halogen full bond breaking strategy of bromotrichloromethane for the straightforward generation of a series of 3-benzoyl-quinolin-2(1*H*)-one acetates with acceptable yields. The further transformation of these resultant 1,5-dicarbonyls into 6/6/6 nitrogen-containing tricyclic skeleton demonstrates more opportunities for applying this policy to polycyclic molecules. This four-component reaction features bond-forming efficiency, broad functional group compatibility and mild reaction conditions. Despite great progress made in this reaction, there are still disadvantages of this synthetic method, such as long reaction time, low atomic economy and cannot be used for late-stage modification of pharmaceutical compounds. Further research of this amide-linked 1,7-diyne is currently being conducted in our group.

## Data Availability

The data presented in this study are available in article and Appendix A.

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
