# Peer review of "Photocatalytic Multicomponent Annulation of Amide-Anchored 1,7-Diynes Enabled by Deconstruction of Bromotrichloromethane"

_molecules, 2024, doi:10.3390/molecules29040782_

Round 1

Reviewer 1 Report

Comments and Suggestions for Authors

The present paper investigates a promising approach using tetrahalomethanes to add one extra carbon into functionalized organic molecules in the aim of form skeletally diverse homologous compounds on some examples. The presented method carries a very high importance in synthetic organic chemistry and proved to be especially promising for generating C1 synthons according to the reported study. The experimental methods are well established, the discussion is supported by the results and is properly concise. Moreover, some mechanistic investigations were also carried out. My overall impression is that the present manuscript is a valuable, well-written one. There are only some minor issues, mainly some missing information, which have to be introduced in a more detailed way before acceptance for publication. These are as follows:

1)Attributed to the high molecular weight of the applied halogenated reagents as C1 sources, I consider the reported synthetic method not so efficient from an atom economic aspect. I suggest the authors to include an additional discussion dedicated for evaluating the results from this aspect.

2)Table 1. The reactions of optimization were carried out at 60 °C in oil bath. Does this temperature value refer to the temperature of the reaction mixture or that of the external medium?

3)The format of the Celsius units is strange, probably contains inappropriate character.

4)The reaction reported in Table 1 are evaluated based on isolated yields. Please supplement the discussion with additional information regarding to the conversion and possible byproducts. How about the conversion after 36 h? Was the formation of any byproducts observed or were there any byproducts isolated?

5) On the basis of which considerations were the applied catalysts selected? Please introduce them in the manuscript. What about other possible alternatives?

6)In the proposed mechanism, the base has one of the roles to deprotonate EtOH. However, many of the applied weak bases are not strong enough to deprotonate the alcohol. Please clarify this topic.

7)In conclusion section the authors wrote “chemical yields”. It is a bit confusing, as there was no mention of other types of yields.

8)In conclusion: “for applying this policy to complex molecules”. Please clarify it and refer to the target groups of future development and further opportunities.

9)The supplement with a short discussion about the limitations or disadvantages of the proposed synthetic method would be welcome. If compared to the traditional C1-introducing procedures, the authors highlighted some less favorable conditions, it would not detract anything from the value of the present work.

After incorporating my minor suggestions, I can recommend the present manuscript for publication.

Comments on the Quality of English Language

English language is fine, only some minor corrections can be considered.

Author Response

The present paper investigates a promising approach using tetrahalomethanes to add one extra carbon into functionalized organic molecules in the aim of form skeletally diverse homologous compounds on some examples. The presented method carries a very high importance in synthetic organic chemistry and proved to be especially promising for generating C1 synthons according to the reported study. The experimental methods are well established, the discussion is supported by the results and is properly concise. Moreover, some mechanistic investigations were also carried out. My overall impression is that the present manuscript is a valuable, well-written one. There are only some minor issues, mainly some missing information, which have to be introduced in a more detailed way before acceptance for publication. These are as follows:

1) Attributed to the high molecular weight of the applied halogenated reagents as C1 sources, I consider the reported synthetic method not so efficient from an atom economic aspect. I suggest the authors to include an additional discussion dedicated for evaluating the results from this aspect.

Response: Thanks for this reviewer’s useful comments. We agree with viewpoint that this reaction has lower atomic economy, and we have added discussion in the summary section.

2) Table 1. The reactions of optimization were carried out at 60 °C in oil bath. Does this temperature value refer to the temperature of the reaction mixture or that of the external medium?

Response: Thanks for this reviewer’s useful comments. This temperature value refers to the temperature of the external oil bath.

3) The format of the Celsius units is strange, probably contains inappropriate character.

Response: Thanks for this reviewer’s useful comments. We have carefully checked and it may be due to typeface issues.

4) The reaction reported in Table 1 are evaluated based on isolated yields. Please supplement the discussion with additional information regarding to the conversion and possible byproducts. How about the conversion after 36 h? Was the formation of any byproducts observed or were there any byproducts isolated?

Response: Thanks for this reviewer’s useful comments. We have added more discussion in the screening of reaction condition section. Unfortunately, we did not isolate and identify any byproducts, except for intermediate 4.

5) On the basis of which considerations were the applied catalysts selected? Please introduce them in the manuscript. What about other possible alternatives?

Response: Thanks for this reviewer’s useful comments. fac-Ir(ppy)3 is an easily available and commercially available metal photocatalyst in the field of organic photocatalysis. We selected this catalyst based on references (Org. Lett. 2021, 23, 7845-7850; Chem. Commun. 2021, 57, 1911-1914) and also screened other iridium catalysts such as [Ir (dFCF3ppy)2dtbbpy]PF6, which have been added to the manuscript.

6) In the proposed mechanism, the base has one of the roles to deprotonate EtOH. However, many of the applied weak bases are not strong enough to deprotonate the alcohol. Please clarify this topic.

Response: Thanks for this reviewer’s useful comments. Based on the reviewer's suggestions, we have revised the reaction mechanism for this step in the manuscript. Intermediate H undergoes nucleophilic 1,6-addition with ethanol, following by proton transfer and removal of HCl to generate intermediate K.

7) In conclusion section the authors wrote “chemical yields”. It is a bit confusing, as there was no mention of other types of yields.

Response: Thanks for this reviewer’s useful comments. We have revised.

8) In conclusion: “for applying this policy to complex molecules”. Please clarify it and refer to the target groups of future development and further opportunities.

Response: Thanks for this reviewer’s useful comments. This sentence refers to this strategy that can be used to construct cyclic molecular frameworks such as compound 3.

9) The supplement with a short discussion about the limitations or disadvantages of the proposed synthetic method would be welcome. If compared to the traditional C1-introducing procedures, the authors highlighted some less favorable conditions, it would not detract anything from the value of the present work.

Response: Thanks for this reviewer’s useful comments. We have added a short discussion about the disadvantages of this synthetic method in the summary section.

Reviewer 2 Report

Comments and Suggestions for Authors

In their manuscript Wang, Zhang and coworkers describe radical-addition triggered cyclization involving triple bond of 2-alkynyl anilides of propiolic acid. Intermediate haloalkenes underwent hydrolysis to produce corresponding carbonyl compounds (ketones and ester). Previous similar reports include cyclisations of ortho-alkynyl-homopropargylbenzenes and o-alkynylaryl propargyl amines or ethers. Therefore, the presented reaction of amides shows enough novelty. Rather good yields are achieved, especially considering the multistep nature of the process. However, structures of some products should be checked (see below for NMR inconsistencies). Thus, the Reviewer can recommend the manuscript for publication in Molecules after some revision.

Major issues are:

-        Optimization table. Did the Author try to perform the reaction at lower temperatures. E.g. at r.t.?

-        Line 114. Please, mention which ortho-substituted and strong EWG-substituted substrates were used.

-        Title should be revised. 1) It is too general. It is not about diynes, but about particular type of diynes – 2-alkynyl anilides of propiolic acid. 2) “perhalogenated methanes” is misleading, since only CBrCl3 was used.

-        Scheme 5. Step C->D. Could this step proceed with CBrCl3 (instead of bromine radical) producing new CCl3 radical? In other words, is the described reaction photocatalytic or photoinitiated?

-        Regarding Scheme 2. Products 2v and 2w. It is interesting that the Authors performed successful photocatalytic reaction with thiols. Usually compounds with RSH moiety are good donors of hydrogen-radical, thus inhibiting radical processes. Could the Authors comment on their success in more detail? Why did CCl3-radical react with substrate and not with RSH: RSH +CCl3-radical = RS-radical + CHCl3.

-        Scheme 2. Could the reaction proceed with R2 other than substituted benzyls, e.g. R2=Me?

-        Figure 1 should be presented more accurate. Now the Authors did not report the conditions, i.e. no concentration of substrates. Moreover all plots were made for different µL values. What was measured in µL? CBrCl3 or its solution? If solution, what was the concentration of this solution?

-        Scheme 5. Did the Authors try the reaction without a base in order to obtain intermediate D?

-        Scheme 5. The Authors show several nucleophilic addition steps: D->E, 4->F, and H->I. Why two of them proceeded with a hydroxide anion, and one of them with ethoxide-anion? Why EtO-anion did not attack intermediates D or 4? Why did hydrolysis of H to corresponding acid not occur?

-        Scheme 5. Could intermediate H convert into a ketene following by the addition of alcohol? Could the Authors comment on that?

-        Product 2l. There are three CH2-groups and one Me-group. However, 5 signals are observed in 13C NMR in 0-100 pm region. Similar compounds 2k and 2m (Hal = F or Br) have similar spectra, while 2l (Hal = Cl) differs. Please, check.

-        Product 2t. The are 14 aromatic protons in the structure. The Author report total integral ratio 13H for region 7.2-8.1 ppm. Please, check.

-        Product 2t. There are four CH2-groups and two acetylenic carbons. However, 7 signals are observed in 13C NMR in 0-100 ppm region. Please, check.

-        Product 2u. There are three CH2-groups. However, 4 signals are observed in 13C NMR in 0-100 pm region. 35 ppm is CH2CO2, 46 is N-CH2Ph, 67 is OCH2Ph. What is 53 ppm? Please, check.

-        Product 2v. There are three CH2-groups and one Me-group. However, 5 signals are observed in 13C NMR in 0-100 pm region. Please, check.

-        Product 2w. There are two CH2-groups, one CH-group, and two Me-group (possibly equal). 4, maximum 5 signals should be in 0-100 ppm region. However, 7 (!) signals are observed in 13C NMR. Please, check.

Minor issues and remarks:

-        The use of term “C1 synthon” is questionable. “C1” does not mean that reagent acts as a precursor of “naked” carbon atom. It means that reagent acts for the installation of one carbon atom into the target molecule (regardless of non-carbon substituents).

-        What is SN’’-substitution? The term is uncommon.

-        Line 81. Why do the Authors call their target product 2a “unexpected”?

-        Scheme 2. Denoting X-substituted benzyls as XBn is incorrect. Benzyl is C6H5CH2. XC6H5CH2 is senseless. Please, revise.

-        Line 134. 2p-2u should be bold.

-        Scheme 4c. Please, provide the percentage of 18O in water and in the product.

-        Scheme 4e. Please, mention reaction temperature and time.

-        Figure 1. Linear fit, not liner fit.

-        Figure 1. BrCCl3, not BrCCL3

-        Scheme 5. Substrate 1a is missed.

-        Scheme 5. Asterisk * for excited state should be placed after the formula, not after the fac.

-        Scheme 5. In “Ir(ppy)3” oxidation state of iridium should be denoted with Roman numerals (e.g. Ir(IV) or IrIV. Alternatively, charge of the whole specie can be indicated in the end (e.g. [Ir(ppy)3]+.)

-        Presentation of DOI looks incorrect. It should be reported as (for example) doi.org/10.1021/acs.orglett.1c02865. or DOI: 10.1021/acs.orglett.1c02865.

Comments on the Quality of English Language

-        The language should be polished. Some of the odd phrases are: “they are abundance and availability”, “perhalogenated methanes (CBrCl3 34 or CBr4) is also widely pursued”, “isocyclic and heterocyclic targets”, “The treatment of 1a and CBrCl3 was irradiated”, “let to extremely poor”, “the influence of different the electronic properties”.

Author Response

In their manuscript Wang, Zhang and coworkers describe radical-addition triggered cyclization involving triple bond of 2-alkynyl anilides of propiolic acid. Intermediate haloalkenes underwent hydrolysis to produce corresponding carbonyl compounds (ketones and ester). Previous similar reports include cyclisations of ortho-alkynyl-homopropargylbenzenes and o-alkynylaryl propargyl amines or ethers. Therefore, the presented reaction of amides shows enough novelty. Rather good yields are achieved, especially considering the multistep nature of the process. However, structures of some products should be checked (see below for NMR inconsistencies). Thus, the Reviewer can recommend the manuscript for publication in Molecules after some revision.

Response: Thanks for this reviewer’s high comments.

Major issues are:

-Optimization table. Did the Author try to perform the reaction at lower temperatures. E.g. at r.t.?

Response: Thanks for this reviewer’s useful comments. The reaction was completely suppressed at room temperature.

-Line 114. Please, mention which ortho-substituted and strong EWG-substituted substrates were used.

Response: Thanks for this reviewer’s useful comments. We have revised.

-Title should be revised. 1) It is too general. It is not about diynes, but about particular type of diynes – 2-alkynyl anilides of propiolic acid. 2) “perhalogenated methanes” is misleading, since only CBrCl3 was used.

Response: Thanks for this reviewer’s useful comments. We have revised.

-Scheme 5. Step C->D. Could this step proceed with CBrCl3 (instead of bromine radical) producing new CCl3 radical? In other words, is the described reaction photocatalytic or photoinitiated?

Response: Thanks for this reviewer’s useful comments. Combined with previous related works (Adv. Synth. Catal. 2022, 364, 2666-2672), we think that this is a photocatalytic process, but it cannot be ruled out that CBrCl3 generates new CCl3 radicals in this step, we will continue further study on this aspect by conducting more experiments and by collaborating with computational chemists.

-Regarding Scheme 2. Products 2v and 2w. It is interesting that the Authors performed successful photocatalytic reaction with thiols. Usually compounds with RSH moiety are good donors of hydrogen-radical, thus inhibiting radical processes. Could the Authors comment on their success in more detail? Why did CCl3-radical react with substrate and not with RSH: RSH + CCl3-radical = RS-radical + CHCl3.

Response: Thanks for this reviewer’s comments. RSH is often used as a hydrogen atom transfer (HAT) reagent. RS- undergoes single electron transfer (SET) with excited state photocatalysts to obtain RS-radicals, which then undergo HAT with the substrate to produce RSH and radical species, which then participate in subsequent reactions instead of directly producing hydrogen radicals (for example: Angew. Chem. Int. Ed. 2019, 58, 312-316). In this reaction, RSH serves as a nucleophilic reagent and reaction medium, and the process does not involve RS-radicals and hydrogen radicals.

-Scheme 2. Could the reaction proceed with R2 other than substituted benzyls, e.g. R2=Me?

Response: Thanks for this reviewer’s useful comments. We have tried to conduct this reaction by using N-Me protected 1,7-diyne 1p as a substrate, but the catalytic system showed complicated products. We will continue this project under new systems following this advice.

-Figure 1 should be presented more accurate. Now the Authors did not report the conditions, i.e. no concentration of substrates. Moreover, all plots were made for different µL values. What was measured in µL? CBrCl3 or its solution? If solution, what was the concentration of this solution?

Response: Thanks for this reviewer’s comments. We have carefully checked and detailed experimental results were displayed in SI.

-Scheme 5. Did the Authors try the reaction without a base in order to obtain intermediate D?

Response: Thanks for this reviewer’s useful comments. The product 2a was observed with 48% yield in the absence of base (Table 1, entry 1). However, intermediate D was not obtained, possibly due to the unstable structure of the conjugated diene of intermediate D.

-Scheme 5. The Authors show several nucleophilic addition steps: D->E, 4->F, and H->I. Why two of them proceeded with a hydroxide anion, and one of them with ethoxide-anion? Why EtO-anion did not attack intermediates D or 4? Why did hydrolysis of H to corresponding acid not occur?

Response: Thanks for this reviewer’s useful comments. Under base conditions, the nucleophilic ability of hydroxide anions is stronger than that of ethanol anions. We cannot provide a clear explanation for why H has not been hydrolyzed into the corresponding acid, possibly due to the presence of a large amount of ethanol in the system. We will continue further research based on this advice.

-Scheme 5. Could intermediate H convert into a ketene following by the addition of alcohol? Could the Authors comment on that?

Response: Thanks for this reviewer’s useful comments. We are unable to separate and characterize intermediate H in this system, we will continue further research based on this advice.

-Product 2l. There are three CH2-groups and one Me-group. However, 5 signals are observed in 13C NMR in 0-100 pm region. Similar compounds 2k and 2m (Hal = F or Br) have similar spectra, while 2l (Hal = Cl) differs. Please, check.

Response: Thanks for this reviewer’s useful comments. We have revised.

-Product 2t. The are 14 aromatic protons in the structure. The Author report total integral ratio 13H for region 7.2-8.1 ppm. Please, check.

Response: Thanks for this reviewer’s useful comments. We have revised.

-Product 2t. There are four CH2-groups and two acetylenic carbons. However, 7 signals are observed in 13C NMR in 0-100 ppm region. Please, check.

Response: Thanks for this reviewer’s useful comments. We have revised.

-Product 2u. There are three CH2-groups. However, 4 signals are observed in 13C NMR in 0-100 pm region. 35 ppm is CH2CO2, 46 is N-CH2Ph, 67 is OCH2Ph. What is 53 ppm? Please, check.

Response: Thanks for this reviewer’s useful comments. We have revised.

-Product 2v. There are three CH2-groups and one Me-group. However, 5 signals are observed in 13C NMR in 0-100 pm region. Please, check.

Response: Thanks for this reviewer’s useful comments. We have revised.

-Product 2w. There are two CH2-groups, one CH-group, and two Me-group (possibly equal). 4, maximum 5 signals should be in 0-100 ppm region. However, 7 (!) signals are observed in 13C NMR. Please, check.

Response: Thanks for this reviewer’s useful comments. We have revised.

Minor issues and remarks:

-The use of term “C1 synthon” is questionable. “C1” does not mean that reagent acts as a precursor of “naked” carbon atom. It means that reagent acts for the installation of one carbon atom into the target molecule (regardless of non-carbon substituents).

Response: Thanks for this reviewer’s useful comments. Based on the references and previous work (Chem. Soc. Rev., 2020, 49, 9197-9219; iScience 2019, 19, 1-13), we believe that it is appropriate to express it as the C1 synthon via quadruple cleavage of carbon-halogen bonds of BrCCl3.

-What is SN”-substitution? The term is uncommon.

Response: Thanks for this reviewer’s useful comments. The meaning is SN2 substitution.

-Line 81. Why do the Authors call their target product 2a “unexpected”?

Response: Thanks for this reviewer’s useful comments. Our expected possible product is compound 4, so product 2 is unexpected.

-Scheme 2. Denoting X-substituted benzyls as XBn is incorrect. Benzyl is C6H5CH2. XC6H5CH2 is senseless. Please, revise.

Response: Thanks for this reviewer’s useful comments. We have revised.

-Line 134. 2p-2u should be bold.

Response: Thanks for this reviewer’s useful comments. We have corrected.

-Scheme 4c. Please, provide the percentage of 18O in water and in the product.

Response: Thanks for this reviewer’s useful comments. We have provided.

-Scheme 4e. Please, mention reaction temperature and time.

Response: Thanks for this reviewer’s useful comments. We have revised.

-Figure 1. Linear fit, not liner fit.

Response: Thanks for this reviewer’s useful comments. We have revised.

-Figure 1. BrCCl3, not BrCCL3

Response: Thanks for this reviewer’s useful comments. We have revised.

-Scheme 5. Substrate 1a is missed.

Response: Thanks for this reviewer’s useful comments. We have revised.

-Scheme 5. Asterisk * for excited state should be placed after the formula, not after the fac.

Response: Thanks for this reviewer’s useful comments. We have revised.

-Scheme 5. In “Ir(ppy)3” oxidation state of iridium should be denoted with Roman numerals (e.g. Ir(IV) or IrIV. Alternatively, charge of the whole specie can be indicated in the end (e.g. [Ir(ppy)3]+.)

Response: Thanks for this reviewer’s useful comments. We have revised.

-Presentation of DOI looks incorrect. It should be reported as (for example) doi.org/10.1021/acs.orglett.1c02865. or DOI: 10.1021/acs.orglett.1c02865.

Response: Thanks for this reviewer’s useful comments. We have revised.

-The language should be polished. Some of the odd phrases are: “they are abundance and availability”, “perhalogenated methanes (CBrCl3 or CBr4) is also widely pursued”, “isocyclic and heterocyclic targets”, “The treatment of 1a and CBrCl3 was irradiated”, “let to extremely poor”, “the influence of different the electronic properties”.

Response: Thanks for this reviewer’s useful comments. We have revised.

Reviewer 3 Report

Comments and Suggestions for Authors

The paper presented by Wang & Zhang describe the first example of visible-light-mediated multicomponent annulation of 1,7- diynes by taking advantage of quadruple cleavage of carbon-halogen bonds of BrCCl3 to generate a C1 synthon. This work is a logical extension of previously groups work. The manuscript is well written with high accuracy. I recommend to publish the article. However, several grammarly and stylish issues should be taken into account.

1. Line 74 "Herein, we document these interesting preliminary results". What does it mean as prelimenary? This paper looks like finished results. Please explain or rewrite the sentence.

2.  Line 39. "....fabricate synthetically...." please replace the 'synthesize'

3. Throughout the article "3-benzoyl-quinolin-2(1H)-one" and similar the H should be in intalic ("H")

4. Line 134. "2p-2u" the bold in missing

5. Line 222. "Funding" has a yellow filling, please check.

6. References. In all references Authours wrote "doi: org/10.1021/cr4002758" (example 1 and so on), this link is not working. It should be like this:  "https://doi.org/10.1021/cr4002758" or this "DOI: 10.1021/cr4002758". Please check and fix in every point of references list.

Author Response

The paper presented by Wang & Zhang describe the first example of visible-light-mediated multicomponent annulation of 1,7- diynes by taking advantage of quadruple cleavage of carbon-halogen bonds of BrCCl3 to generate a C1 synthon. This work is a logical extension of previously groups work. The manuscript is well written with high accuracy. I recommend to publish the article. However, several grammarly and stylish issues should be taken into account.

Response: Thanks for this reviewer’s high comments.

  1. Line 74 "Herein, we document these interesting preliminary results". What does it mean as prelimenary? This paper looks like finished results. Please explain or rewrite the sentence.

Response: Thanks for this reviewer’s useful comments. We used inappropriate words and have already revised this sentence.

  1. Line 39. "....fabricate synthetically...." please replace the 'synthesize'

Response: Thanks for this reviewer’s useful comments. We have replaced.

  1. Throughout the article "3-benzoyl-quinolin-2(1H)-one" and similar the H should be in intalic ("H")

Response: Thanks for this reviewer’s useful comments. We have corrected.

  1. Line 134. "2p-2u" the bold in missing

Response: Thanks for this reviewer’s useful comments. We have corrected.

  1. Line 222. "Funding" has a yellow filling, please check.

Response: Thanks for this reviewer’s useful comments. We have carefully checked and revised.

  1. References. In all references Authours wrote "doi: org/10.1021/cr4002758" (example 1 and so on), this link is not working. It should be like this: "https://doi.org/10.1021/cr4002758" or this "DOI: 10.1021/cr4002758". Please check and fix in every point of references list.

Response: Thanks for this reviewer’s useful comments. We have revised.

Round 2

Reviewer 2 Report

Comments and Suggestions for Authors

The manuscript appears much better after the revision. However, there are still some issues.

-        Title. Bromotrichloromethane, not Bromotrichloromethanes.

-        Denote abbreviation “Ir(dFCF3ppy)2dtbbpy]PF6”

-        Mes-Acrl+ClO4- in the text, but Mes-Acr+ClO4- in the table. Please, check.

-        When reporting doi, there should be no space between “doi.” and “org”.

-        Concerning step C->D, the Authors said in response to Reviewer: “it cannot be ruled out that CBrCl3 generates new CCl3 radicals in this step”. Possibly, it should be also mentioned in the manuscript, not only in the response.

-        Concerning the reaction with thiols, the Reviewer is not fully satisfied with the response. No additional information was added to the manuscript. In J. Org. Chem. 2013, 78, 2046 (doi: 10.1021/jo3020825) one can see that RSH can produce thiyl (RS) radical in the absence of base. In Tetrahedron 2014, 70, 4264 (doi: 10.1016/j.tet.2014.03.041) C. Stephenson proposed CBrCl3 as a radical initiator for thiol-ene reaction. Why all such processes did not interfere with the diyne-cyclization reported by the Authors?

-        Figure 1 should be presented more accurately. The reader should not necessarily look at SI to understand the figure. Some more data should be added (see report from the 1st stage of revision)

-        (see stage 1 of the revision). Scheme 4c. Please, provide the percentage of 18O (with respect to 16O) in water and in the product.

-        Title in the SI was not revised.

Comments on the Quality of English Language

Some unclear phrases are:

-        “to fabricate synthesize”, “raw material” (starting materials?), “no by-products are separated” (is it about side products?), “electron withdrawing groups (p-Br, p-CO2Me) were suppressed during the reaction process” (what was suppressed? Groups or reaction?), “could carry out smoothly”, “Benzyl benzene ring”, “was an unreactive reactant.

Author Response

-Title. Bromotrichloromethane, not Bromotrichloromethanes.

Response: Thanks for this reviewer’s useful comments. We have revised.

-Denote abbreviation “Ir(dFCF3ppy)2dtbbpy]PF6”

Response: Thanks for this reviewer’s useful comments. We have added the structural formulas of all photocatalysts in Table 1.

-Mes-Acrl+ClO4- in the text, but Mes-Acr+ClO4- in the table. Please, check.

Response: Thanks for this reviewer’s useful comments. We have revised.

-When reporting doi, there should be no space between “doi.” and “org”.

Response: Thanks for this reviewer’s useful comments. We have revised.

-Concerning step C->D, the Authors said in response to Reviewer: “it cannot be ruled out that CBrCl3 generates new CCl3 radicals in this step”. Possibly, it should be also mentioned in the manuscript, not only in the response.

Response: Thanks for this reviewer’s useful comments. Although we believe that the possibility of CBrCl3 directly generating new CCl3 radicals in this step cannot be ruled out, in the first step of the reaction, Ir (III)* species reacts with BrCCl3 to produce CCl3 radicals via single electron transfer (SET), indicating that photocatalysts are necessary. If we mention this process in the manuscript, it will be contradictory and may cause misunderstandings for readers. The proposed reaction mechanism is reasonable and supported by literature. Thus, this process will not be mentioned in the manuscript.

-Concerning the reaction with thiols, the Reviewer is not fully satisfied with the response. No additional information was added to the manuscript. In J. Org. Chem. 2013, 78, 2046 (doi: 10.1021/jo3020825) one can see that RSH can produce thiyl (RS) radical in the absence of base. In Tetrahedron 2014, 70, 4264 (doi: 10.1016/j.tet.2014.03.041) C. Stephenson proposed CBrCl3 as a radical initiator for thiol-ene reaction. Why all such processes did not interfere with the diyne-cyclization reported by the Authors?

Response: Thanks for this reviewer’s useful comments. We regret that we are unable to provide a satisfactory response to the reviewers. We propose possible reaction mechanisms based on the experimental results. The experimental results are inconsistent with the literature (J. Org. Chem. 2013, 78, 2046; Tetrahedron 2014, 70, 4264), which is the charm of chemistry, interesting and unexpected. If RS radicals are generated in the system, they may also be quickly quenched to produce RSH because intramolecular reactions are faster than intermolecular reactions. We will continue this project under the new system to achieve the cyclization of diynes induced by thiol radicals.

-Figure 1 should be presented more accurately. The reader should not necessarily look at SI to understand the figure. Some more data should be added (see report from the 1st stage of revision)

Response: Thanks for this reviewer’s useful comments. We have revised. We have carefully checked again and do not understand what data needs to be provided? The substrate concentration was provided in SI; µL values measure 1a or CBrCl3 (1.0× 10-4 mol/L), due to the small amount of added solution (1a or CBrCl3), the volume change is negligible. We have described in detail and are consistent with the references.

-(see stage 1 of the revision). Scheme 4c. Please, provide the percentage of 18O (with respect to 16O) in water and in the product.

Response: Thanks for this reviewer’s useful comments. We have provided the percentage of 18O (with respect to 16O) in water and in the product. Due to the Chinese Spring Festival, our school's analysis and testing center is on vacation, and we are unable to provide an enlarged HR-MS image of product 2a. However, the overall HR-MS image and existing data can also reflect the ratio of 16O and 18O.

-Title in the SI was not revised.

Response: Thanks for this reviewer’s useful comments. We have revised.

Comments on the Quality of English Language

Some unclear phrases are:

-“to fabricate synthesize”, “raw material” (starting materials?), “no by-products are separated” (is it about side products?), “electron withdrawing groups (p-Br, p-CO2Me) were suppressed during the reaction process” (what was suppressed? Groups or reaction?), “could carry out smoothly”, “Benzyl benzene ring”, “was an unreactive reactant.

Response: Thanks for this reviewer’s useful comments. We have revised.